# Reinforcement learning in cold atom experiments

Malte Reinschmidt[1], József Fortágh[1], Andreas Günther[1] ✉ & Valentin V. Volchkov [2] ✉

Cold atom traps are at the heart of many quantum applications in science and technology. The preparation and control of atomic clouds involves complex optimization processes, that could be supported and accelerated by machine learning. In this work, we introduce reinforcement learning to cold atom experiments and demonstrate a flexible and adaptive approach to control a magneto-optical trap. Instead of following a set of predetermined rules to accomplish a specific task, the objectives are defined by a reward function. This approach not only optimizes the cooling of atoms just as an experimentalist would do, but also enables new operational modes such as the preparation of pre-defined numbers of atoms in a cloud. The machine control is trained to be robust against external perturbations and able to react to situations not seen during the training. Finally, we show that the time consuming training can be performed in-silico using a generic simulation and demonstrate successful transfer to the real world experiment.

Laser cooling and magneto-optical trapping of neutral atoms has been one of the major breakthroughs in science in the last decades[1]. It enabled access to ultracold atoms and quantum matter with many applications in quantum simulation, computation, and sensing[2]. Nowadays, the magneto-optical trap (MOT) is the cornerstone for all applications and devices based on ultracold atoms[3]. Despite its simple design, operation can be quite intricate, with the behavior of the atoms being determined by the interplay of laser cooling, optical pumping, and light-induced interactions[4–6]. With the advent of non-alkali and molecular MOTs[7–9] also the number of lasers used is rising. This increasing complexity, albeit providing major scientific advances, can create difficulties in controlling and optimizing the production of ultracold samples. Handling and making use of this complexity, as well as developing MOT-based devices that are robust outside of lab conditions, are the emerging key challenges. Machine learning and algorithmic optimization constitute increasingly popular and effective approaches to improving control sequences of different phases of an experiment and even creating a BEC. One general approach is to parameterize the ultracold atom experiment via a vector of experimental settings such as currents, voltages, and durations. For each experimental parameter setting a cost is defined as a function of the experiment's outcome. The goal of the optimization is then to find the parameter settings corresponding to the global optimum of the cost. For this, the cost is typically approximated by Gaussian processes[10,11] or artificial neural networks[12–14]. The optimum in the parameter space is then obtained using algorithms such as Bayesian optimization[15], evolutionary optimization[13], or 'citizen optimization'[16], all showing significant improvement of the performance of the experiments compared to manual optimization. However, these strategies yield a fixed vector of parameter settings and cannot react to perturbations during an experimental cycle or deal with possible long term drifts of experimental conditions. Instead, optimization needs to be repeated every time the conditions change.

These challenges can be mitigated using active feedback control, which in cold atom systems could be realized via a conventional feedback loop implemented on an FPGA[17]. However, this requires a precise knowledge of the controlled system and a physical understanding of the observables as well as highly task-specific manual tuning of the controller.

Alternatively, the experimental control can be formulated as a sequential decision-making problem. It consists of dividing the experimental cycle into time steps, so that during the execution of an

[1]Center for Quantum Science, Physikalisches Institut, Eberhard Karls Universität Tübingen, Tübingen, Germany. [2]Max Planck Institute for Intelligent Systems, Tübingen, Germany. ✉e-mail: a.guenther@uni-tuebingen.de; valentin.volchkov@tuebingen.mpg.de

experiment, at each time step the state of the system is observed, and based on that observation a control decision is taken. The choice of approach for sequential decision-making problems is reinforcement learning (RL)[18]. In the last decade, RL has shown human-level performances in a range of tasks that were previously thought to be impossible for machines to complete. While impressive success was achieved in games and simulated environments[19–23], application of RL to real world systems remained challenging[24]. In recent years, control of various physical systems was established through RL algorithms, leading to a paradigm shift in how control is approached and implemented[25–30]. Generally speaking, in RL the control algorithm denoted as *agent* interacts with an environment based on observations, conceptually similar to a classical PID-controller. As a result of its actions the agent receives rewards. Contrary to the cost in traditional optimization, the reward is not optimized directly, but used in a training algorithm, that allows the agent to learn over time how to maximize the cumulative reward. The task is defined only through the reward function, leaving the specific control sequence solely to the agent. So-called deep RL[31] incorporates deep artificial neural networks allowing for model-free control as a function of many, potentially high dimensional inputs, such as images.

Here, we introduce deep RL to cold atom experiments by means of controlling a MOT-system. We enhance an existing cold atom apparatus used to cool rubidium ($^{87}$Rb) atoms by an RL agent, taking advantage of the fluorescence light, which is a by-product of laser cooling. In most similar apparatuses, fluorescence images are only a basic diagnostic tool, whereas they can contain a complex and nonlinear representation of the system. Therefore, we utilize the live fluorescence images as input for the control, i.e., the images act as observations for the RL agent. We use an off-the-shelf implementation[32] of a popular RL algorithm with minimal modifications to the code, running on a desktop computer. As proof-of-concept, we perform training of the agent directly on the experiment, which rapidly converges to a cooling scheme resembling well known molasses cooling. Furthermore, we train the agent to be capable to react to sudden changes in the environment and observe generalization to situations that are not included during the training. As a result, we achieve stability with respect to perturbations within a single sequence, shot-to-shot parameter fluctuations over successive sequences, and long-term drifts of the lab conditions. We demonstrate new modes of operations of the MOT, such as loading predefined number of atoms by engineering the corresponding reward function. Finally, we explore how an agent trained on a simple, neural network-based MOT simulator can be transferred to the real world experiment, opening the avenue to fast pre-training, extensive parameter optimization and large number of control parameters.

## Results

### Experimental system
The demonstration of RL-based MOT control was implemented on a cold atom apparatus, that employed many standard techniques (see "Methods"). Similar setups can be found in many labs around the world (https://everycoldatom.com/). Like all experiments with cold atoms, these experiments were performed in recurring cycles, each starting with the preparation of an atomic cloud in a MOT. We dedicate the present manuscript to this preparation phase and conclude the experimental cycle directly thereafter by detecting the collected atoms. For that, we used time-of-flight absorption imaging[33], in which the expanding cold atom cloud was illuminated with a resonant laser beam, and the beam transmission measured on a CCD camera. This image was used to determine the final state of the atomic cloud, consisting of the number of atoms $N$ and their average kinetic energy, which we denote by 'temperature' $T$. In our specific system, after a loading time of 1.5 s, without reaching saturation the number of atoms typically amounted to $N \approx 1.5 \times 10^8$ at a temperature of about 100 μK.

### Implementation of the control
While most of the experimental control was running on a separate system, the agent algorithm was implemented on a desktop PC with access to live fluorescence images and control over specific experimental parameters (see "Methods"). We limited the parameters controlled by the agent to the frequency detuning $\Delta = \omega_0 - \omega_L$ of the cooling laser at frequency $\omega_L$ with respect to the atomic resonance at $\omega_0$ and the magnetic field gradient $B'$ at the MOT center along the field's symmetry axis (see "Methods"). They are among the most influential parameters, both on the number of atoms and the temperature in the MOT, and are typically varied over the course of the MOT loading. In this study, the RL agent was allowed to continuously adjust one or both of these parameters. The detuning $\Delta$ was restricted to the range of 0–8.25 Γ with Γ = $2\pi \times 6.063$ MHz being the cooling transition linewidth[34]. The magnetic field gradient $B'$ was varied by the agent between 7.5–22.5 G cm$^{-1}$.

Each MOT cycle of 1.5 s duration constituted an episode and was divided into 25 sequential time steps $n_t$ of 60 ms length. Figure 1a, b illustrate the main principle for controlling the MOT with an RL agent and the corresponding temporal sequence, respectively. The MOT was part of the environment, which denotes the entire physical system, including the atoms, the surrounding experimental set-up, and the lab. Within each time step a vector of observables consisting of four most recent fluorescence images and additional parameters such as current time step index and values of control parameters was passed as input to the agent. The agent processed the input using a trainable neural network and determined the next action, i.e., computed the output values for the controlled experimental parameters. Subsequently, the control parameters were applied through the experimental control, and the environment evolved for the duration of the time step. After each time step, the agent stored the observables together with the corresponding reward and action taken into a replay memory.

### Training of the agent
In order to train the agent for a specific task, the reward function needs to be defined accordingly. In general, a reward can be generated after each time step, in our implementation a reward was attributed only after the last time step when the state of the atomic cloud has been obtained through absorption imaging. The essence of RL is to train the agent's policy to maximize the reward. The RL algorithm used here was the deep deterministic policy gradient (DDPG)[35], which has been developed for high dimensional observations (e.g., images) and is capable of outputting continuous actions, i.e., control parameters. The training process consisted in optimizing the policy using the observations and the associated rewards sampled from the replay memory obtained throughout the training period (see "Methods"). An optimization step took place in between the training episodes, as shown in Fig. 1b. During the training phase, exploration was introduced via noise added to the actions of the agent. A single training consisted of up to $10^5$ episodes, depending on the number of control parameters and the specific definition of the reward function. Figure 1b shows a successful training, in the course of which the obtained reward increased and saturated, while the output values (control parameters) converged to a stable sequence (see insets). Figure 1c shows a typical control sequence within one episode, together with representative fluorescence images.

A well known problem of learning algorithms is that some inputs may be ignored and instead "shortcuts" are memorized[36]. In our scenario, this approach was particularly tempting as the experimental setups are typically sufficiently stable and run reproducibly, thus operation with static control sequences is very common. To prevent learning of shortcuts, we introduced a training perturbation, i.e., an offset, that was randomly generated at the start of each episode and added to the agent's output value of the control parameter, effectively shifting the experimental conditions. Such a perturbation is equivalent

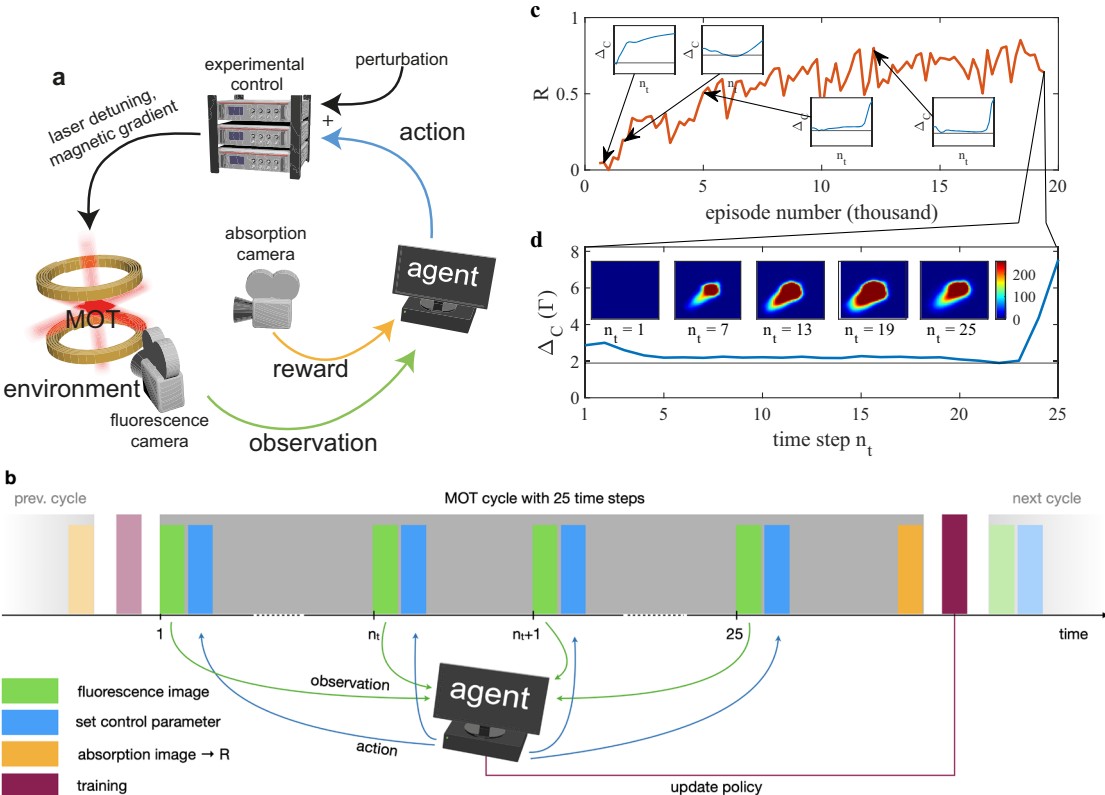

**Fig. 1 | Reinforcement learning for MOT. a** Schematic illustration and **b** timing of the interaction between an RL agent and a MOT-system. The agent takes control (action) via dedicated control parameters, which it freely adjusts during the MOT operation. At each time step $n_t$, the action is based upon the observables (e.g., fluorescence images) given to the agent and the knowledge acquired during the training phase. The latter is based on a reward value $R$, extracted at the end of the MOT cycle (episode) from an absorption image of the atom cloud, which the agent aims to maximize. **c** Evolution of reward $R$ (normalized to reference sequences, as described in the "Methods") during a successful training. The insets show the control parameter sequences at four points during the training phase. Animated fluorescence images corresponding to the episodes in the insets are provided in Supplementary Movie 1. **d** Sequence of control parameter ($\Delta_C$) over the course of an episode for an agent trained to maximize $N/T$. The horizontal line indicates the value for maximal loading rate of the MOT. The inset shows the fluorescence images observed by the agent at various time steps $n_t$ during the episode.

to a shot-to-shot fluctuation of the control parameter that may occur in experiments. As the offset value was unknown to the agent, it was forced to extract information from the fluorescence images in order to compute the control parameter. This approach is similar to *domain randomization* that induces variability during training on simulations, making the agent more robust in real world environments[37].

## Evaluating the agent's performance

In order to evaluate and compare the performance of trained agents, we defined a testing procedure, aiming for the agent's ability to adapt to different experimental conditions, as well as the reproducibility of the results when presented with the same conditions. As such, we tested the agent by adding different predetermined perturbation offset values (unknown to the agent) to the control parameter at the start of an episode, repeating the test for each of these values up to 200 times. We then compared the final results in terms of the number of collected atoms, the achieved cooling, and the corresponding reward. Additionally, we studied to which policy the agent has converged by analyzing the sequences and spread of the output values over the course of multiple episodes for each of the different offset values (Note, that the agent does not differentiate between training or evaluation episodes, since its actions solely depend on the obtained observations).

## Standard MOT operation mode

In many cold atom experiments MOT operation aims for maximizing the number of trapped atoms while reaching the lowest possible

temperature, as it is one of the keys towards quantum degeneracy. For a proof-of-principle demonstration we defined the reward as $R \propto N/T$ and focused first on a single control parameter—the laser detuning $\Delta$. The physical value of the laser detuning $\Delta$ was determined through the agent's control value $\Delta_C$ and the perturbation offset $\Delta_{\text{Offset}}$ (unknown to the agent) via $\Delta = \Delta_C - \Delta_{\text{Offset}}$. The magnetic field gradient was set to $B' = 15 \, \text{G cm}^{-1}$.

With the MOT principles being based on velocity dependent light forces, the loading rate of atoms strongly depends on laser detuning and shows a maximum at a specific value. In our experiment we measured this value to $\Delta_{\text{opt}} = 1.9 \, \Gamma$. The temperature is known to approximately scale inversely proportional to $\Delta$[38]. The final temperature was then mainly given by the value of $\Delta$ applied at the end of the episode. With this in mind, the intuitive optimal policy consisted of first going to the point of highest loading rate, before rapidly ramping $\Delta$ to the maximum value at the end of the episode to reduce the temperature. This policy corresponds to what is commonly used in labs working with MOTs and provides a reference for studying the application of RL to MOT control (see "Methods").

In Fig. 1c, we present the learning progress: the agent discovered early on that going to large detunings at the end of an episode resulted in a massive boost in the reward, as witnessed in the first insets. This is consistent with the fact that reward is issued only in the last time step. During the second half of the training, the agent fine-tuned the loading phase. The control output sequences $\Delta_C$ for a trained agent evaluated at two different detuning offsets are shown in Fig. 2a, b. We observe, that the sequence of the detuning $\Delta_C$ over the course of an episode for

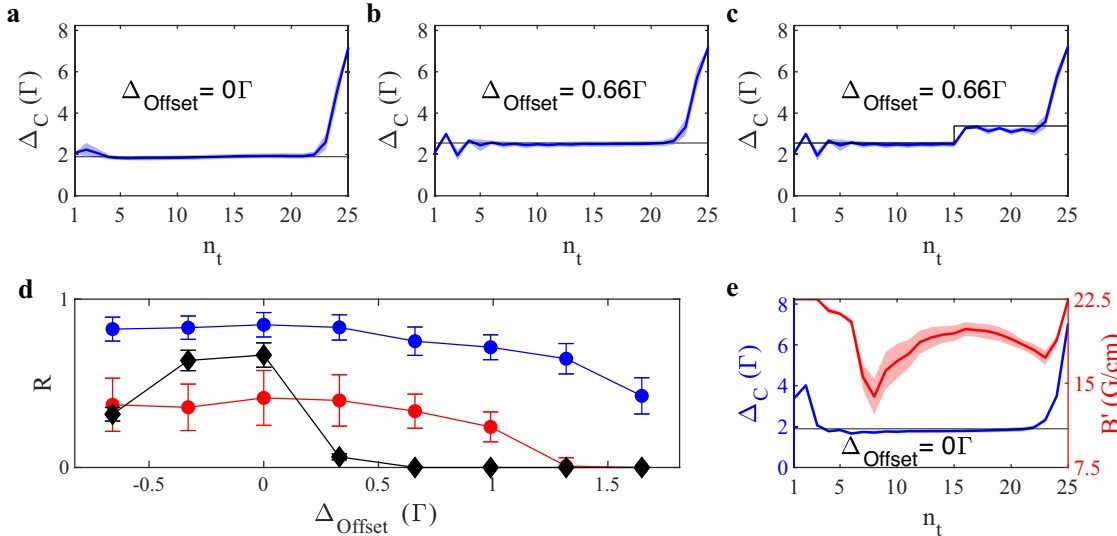

**Fig. 2 | Evaluation of agent trained to optimize $N/T$. a** Control parameter sequence over the course of episodes with mean value (colored solid line) and standard deviation (transparent region). Data are obtained by an agent evaluated at $\Delta_{\text{Offset}} = 0$. The horizontal black line indicates the point of optimal loading $\Delta_{\text{opt}}$. **b** Same agent as in **a**, but evaluated at an offset $\Delta_{\text{Offset}} = 0.66\,\Gamma$. The optimal loading line has been shifted accordingly. **c** Same agent as in **b**, but evaluated with a sudden step in the $\Delta_{\text{Offset}}$ at $n_t = 15$ with amplitude $0.82\,\Gamma$. **d** Reward values (mean and standard deviation) of three different agents as extracted during the evaluation for different $\Delta_{\text{Offset}}$ values: blue circles/red circles—agent controlling perturbed detuning ($\Delta_{\text{Offset}} \neq 0$ during training); black diamonds—agent controlling unperturbed detuning ($\Delta_{\text{Offset}} = 0$ during training); The measurements shown as red circles include a sudden change in the $\Delta_{\text{Offset}}$ value at $n_t = 15$ during evaluation, as illustrated in **c**. The timing and amplitude of this "step" in the offset was the same for all detuning offsets. Error bars correspond to the standard deviation of 200 repetitions of the measurement. **e** Control parameter sequence over the course of episodes for an agent controlling two output parameters, corresponding to the cooling laser detuning $\Delta_C$ (blue) and the magnetic field gradient $B'$ (red).

a trained agent matches our expected intuitive policy. While the first fluorescence images of each episode were blank, as no atoms had yet been loaded into the MOT, the agent developed an initial search pattern, jumping to different values of $\Delta_C$. Once the first atoms were trapped and visible on the fluorescence image, the agent compensated for the perturbation offset $\Delta_{\text{Offset}}$ and set the output value corresponding to the highest loading rate, as shown in Fig. 2b. This allowed the agent to load as many atoms into the MOT as possible, before increasing $\Delta_C$ towards the end of the episode in order to reduce the temperature and maximize the reward. Figure 2d summarizes the performance in terms of the achieved rewards for several trained agents, evaluated at different offsets. At large offsets, the drop in the obtained reward was due to an increased number of time steps needed to locate the optimal loading point.

For sake of consistency, we also trained an agent without perturbations on the control values. This agent learned the optimal policy, however, it ignored the changes related to $\Delta_{\text{Offset}}$ in the observations during the evaluation phase and played back the same memorized action sequence. The reward then simply mirrors the loading rate of the MOT at different offsets from the optimal loading point, as can be seen in Fig. 2d.

### Generalization to unfamiliar situations

The capacity to adequately react to new situations, unseen during the training, is one of the key interests in RL since it is a witness of the robustness and usefulness of the trained agent. Thus, we tested our agent by exposing it to sudden changes in the environment and observing the real-time adaption of its behavior. In particular, the offset applied to the output value $\Delta_C$ during evaluation was suddenly increased at $n_t = 15$ by $0.82\,\Gamma$. Such a shift can be compared to a sudden change in the environment, which may be caused by singular acoustic or electronic events. As shown in the control parameter sequence of Fig. 2c, the agent was able to compensate for this shift after just a single step, i.e., based on a single fluorescence image. This is remarkable, since it demonstrates that the agent has discovered patterns in the

shape of the fluorescence allowing to estimate the current detuning value. This results in the ability of the trained agent to generalize and adapt to conditions not present during the training. Note, that the corresponding reward values (● in Fig. 2d) cannot be directly compared to those without shift (● in Fig. 2d), as for a short interval of time, before the agent was given a chance to react, the detuning was being shifted close enough to the resonance where loading not only stops, but atoms were actively removed from the trap. As a result, the total collected number of atoms was reduced, leading to lower reward values.

### Increasing action space dimension

So far, the action space has been limited to a single control parameter. In the following we expanded the action space to also include the current that defined the gradient of the magnetic field, i.e., $B'$, while the definition of the reward as $R \propto N/T$ and other aspects during the training remained unchanged. The perturbation offset applied to $\Delta$ was found to be sufficient in preventing the learning of shortcuts, so no offset was added to $B'$.

With the controls extended to the gradient of the magnetic field, schemes similar to compressed MOT[4] or polarization gradient cooling[38] could be expected. However, these techniques require control on time scales much shorter than the time step duration of 60 ms in our experiment. As a consequence, no intuitive optimal policy is known for this scenario.

We evaluate the trained agent, similar as before, with the resulting control sequences for zero offset shown in Fig. 2d. The control of the detuning $\Delta$ converged to a similar sequence as described in the previous section, i.e., the agent learned to compensate for the detuning offset. As for $B'$, the agent initially maximized this control parameter. We interpret this as an attempt to increase the sensitivity to fluorescence of small atom numbers in the beginning of an episode. Once the point of highest loading rate was reached, $B'$ was lowered. In a manner reminiscent of a compressed MOT, $B'$ was increased to maximum at the end of the episode.

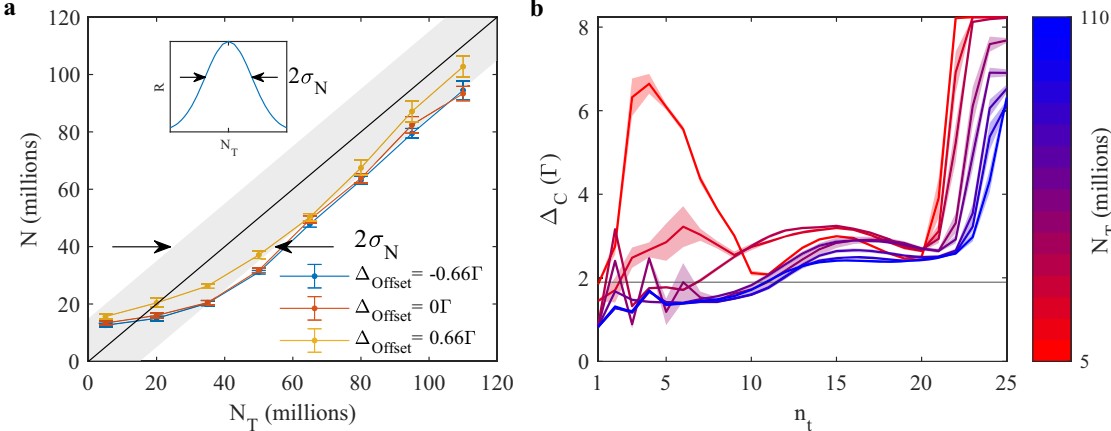

**Fig. 3 | Atom number on demand. a** Evaluation for three different offsets in terms of the targeted ($N_T$) and achieved ($N$) number of atoms for an agent trained to load an adjustable amount of atoms. The inset shows the reward function, defined as gaussian profile around $N_T$, with standard deviation $\sigma_N = 15 \times 10^6$. The black line and gray area correspond to the response $N = N_T \pm \sigma_N$, with mean and standard deviation, respectively. The error bars of the data points correspond to the standard deviation of 21 repetitions of the measurement. **b** Control parameter sequence for $\Delta_{offset} = 0$ and different vales of $N_T$, ranging from $5 \times 10^6$ (red) to $110 \times 10^6$ (blue). Qualitatively similar sequences were observed for non-zero offsets.

All together, it is remarkable that the agent, despite the increased complexity, found a reliable strategy to generate a MOT and compensate for the offset. We expect MOT operation to be improved, provided the agents is granted control over further experimental parameters.

## Reward function engineering

Now we go beyond the standard task of maximizing $N/T$ and showcase how the engineering of the reward function enables the realization of new modes of operation without extending the experimental setup. As an example, one important task is the preparation of a sample with an adjustable number of atoms.

A straightforward solution would be to set a threshold for the integrated fluorescence signal, after which the sequence would stop. However, this approach would fail in the case of drifting detuning. Furthermore, the asynchronicity of the control would result in uncontrolled sequence durations.

Another approach consists in preparing a sample of atoms and subsequently reducing it to a desired atom number via a controlled loss mechanism. For mesoscopic MOT systems, superfluous atoms were ejected using pulses of magnetic gradients or repump beams[39,40]. Larger number-stabilized samples were prepared in a magnetic trap using RF pulses as feedback after non-destructive imaging measurements of the number of atoms[17]. Precise characterization of the imaging process and loss mechanism allowed to obtain sub-shot-noise atom number precision for user specified targets, however, only in a relatively narrow range of 2.5–5 million atoms.

Using RL, we were able to train an agent to achieve variable targeted atom numbers in each episode, while acting only on the detuning of the cooling laser and keeping all other parameters fixed ($B' = 15\,\mathrm{G\,cm^{-1}}$). For this, we introduced a new parameter $N_T$ for the target number of atoms and passed it to the agent as additional observable. The reward function was altered to depend on this parameter, using a Gaussian profile centered around $N_T$ with a width $\sigma_N$. To ensure that the atoms remain cold, we included a factor $1/T$ in the reward function:

$$R \propto \frac{1}{T} \exp\left(-\frac{(N - N_T)^2}{2\sigma_N^2}\right) \qquad (1)$$

For the training we limited the range of $N_T$ to 0–120 million atoms and set the acceptance range $\sigma_N$ equal to 15 million atoms. $N_T$ was randomly generated for each training episode, with a uniform distribution across the entire range. As before, a random perturbation offset was added to the output of the agent at each training episode to prevent the learning of shortcuts. The agent, therefore, had to learn to manipulate the loading rate to achieve the requested atom number while simultaneously compensating the offset.

Intuitively, one possible control strategy could be to divide the episode in two phases: a loading phase with the detuning remaining at the optimal loading point but its length adjusted to meet the requested atom number, followed by a cooling phase at larger detunings. Alternatively, the loading could take place at variable loading rate but constant length and terminated with a jump to maximal detuning for cooling at the episode's end.

We evaluated the trained agent at given $N_T$ values, ranging from 5–110 million atoms. When comparing the measured number of atoms to the target values in Fig. 3a, we find, that they fall roughly within a region of $\pm\sigma_N$ around $N_T$. The agent was thus capable of adapting to different numbers of atoms in consecutive episodes. More importantly, the performance of the agent was robust with respect to the applied perturbation during the evaluation, as demonstrated by the data obtained for non-zero $\Delta_{Offset}$. This robustness is a direct consequence of the real-time feedback of the RL agent and would be impossible to obtain for manual and static control sequences.

The control parameter sequences presented in Fig. 3b were applied by the agent for $\Delta_{Offset} = 0$. They show that the agent varied both: the loading rate (i.e., distance to $\Delta_{opt}$) as well as the duration of the cooling phase (i.e., jump to larger detuning) depending on $N_T$. Furthermore, for increasing values of $N_T$ the agent limited the cooling in a trade-off for larger number of atoms. Indeed, increasing the detuning reduces the confinement of the MOT and leads to an increased loss of atoms. Due to the absolute nature of the acceptance range $\sigma_N$ in the reward function (1), the loss of atoms affects the reward only for larger $N_T$. In principle, the agent could collect a larger number of atoms in order to compensate for this loss and to better meet the target by staying closer to $\Delta_{opt}$ during the loading phase. This shows that despite its overall success, the policy of the trained agent is not yet optimal.

Similar to our approach, in ref. 13 the cost function was tailored to prepare samples at a given temperature, while maximizing the number of atoms. Analogously, the cost function could be adapted to produce samples with a fixed number of atoms,

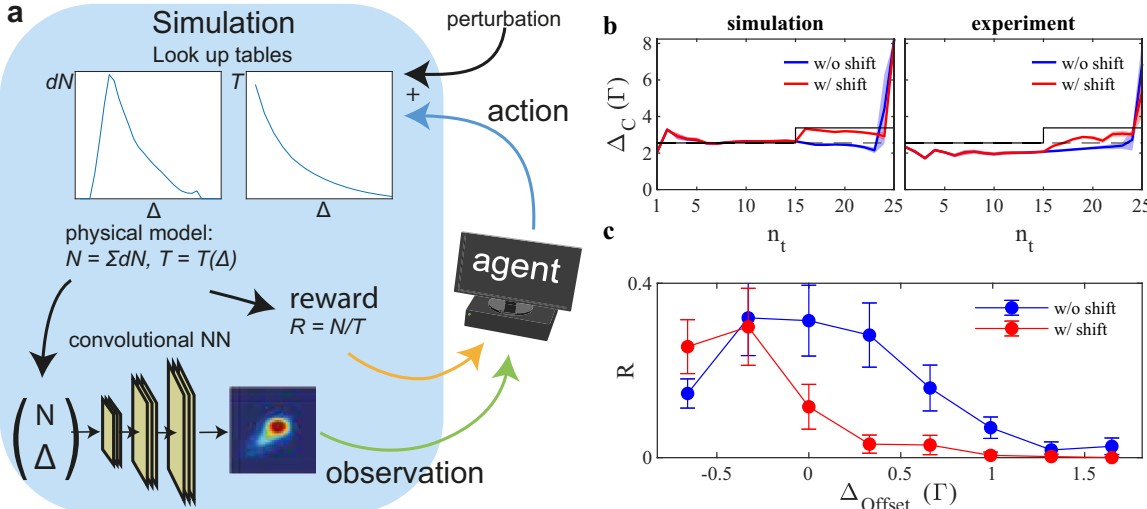

**Fig. 4 | Sim-to-real transfer. a** Schematic diagram of the training on the simulation: the graphs illustrate the experimentally measured values for the number of atoms $dN$ loaded during a time step and the temperature $T$ as function of the detuning $\Delta$. At the bottom the generation of fluorescence images is illustrated: a convolutional neural network takes the number of atoms $N$ and $\Delta$ as input to produce realistically looking images (see "Methods"). **b** Control parameter sequences of an agent trained on the simulation and evaluated for $\Delta_{offset} = 0.66\Gamma$ on the simulation (left)

and experiment (right). The red curves show the resulting behavior if the offset is suddenly changed during the episode (cf. Fig. 2c). The solid/dashed black line indicate the point of optimal loading with/without the sudden change of the offset. **c** Reward values of agents trained with simulation and evaluated on the experiment. blue circles/red circles − reward without/with sudden change of the offset. Error bars correspond to the standard deviation of 200 repetitions of the measurement.

however, changing the target would require retraining of the model from scratch.

## Sim-to-real transfer

As with other implementations of RL in real world environments, a major challenge is the required training time. Depending on the complexity of the reward function and the number of parameters, the training performed for this work took 12−48 h. Adding complexity to the reward function, or expanding the system to include more parameters increases this time.

A typical approach to this problem in RL for robotic systems is sim-to-real transfer. If given a simulation sufficiently close to the environment, general behavior can be studied and hyperparameters of the algorithm being optimized. Ideally, the algorithm trained on the simulation could then be transferred to the experiment, either directly, or as a basis for an additional shorter training to adapt to the real world.

To investigate sim-to-real transfer, we have implemented a simple phenomenological model of the MOT, as depicted in Fig. 4a and detailed in the "Methods" section. We modeled the internal state of the MOT assuming that both, the number of atoms $dN$ loaded per time step as well as the temperature $T$ only depend on the current value of $\Delta$ and extracted their functional relation, $dN(\Delta)$ and $T(\Delta)$, from the experiment. Furthermore, we trained a generative neural network on experimental data to reproduce fluorescence images corresponding to the simulated MOT state. As with the training on the experiment, a perturbation offset was applied to the action in order to force the agent to rely on the fluorescence images.

Figure 4b (left) shows the control sequence of an agent trained and evaluated on the simulated environment, reproducing the behavior of experimentally trained agents. We then evaluated the simulation-trained agent directly on the real world experiment. The corresponding control presented in Fig. 4b (right) qualitatively mimics the behavior of the agent evaluated on the simulation and thereby indicates a successful sim-to-real transfer. More importantly, when presented with a sudden shift of the offset value during the episode,

the sim-to-real agent adapted to the change in the same manner as the agent trained on the experiment. The agent was, therefore, not only able to generalize from its training on the simulations to the real world experiment, but was even able to take this a step further and generalize to unknown situations that did not occur during the training.

The absolute reward values achieved by the sim-to-real agent, shown in Fig. 4c, are inferior to those of the agents trained directly on the experiment. The main reason for this is that the agent did not fully compensate for the perturbation offset when evaluated on the experiment, as can be seen in Fig. 4b. This shows that there is still a considerable gap between the simulations and the experiments. Possibly, one contributing factor is the smoothing of the synthetic images, lacking high spatial frequency features of the fluorescence images in the experiment (see "Methods"). Other factors may be the simplifications in the simulation, which ignore light-induced interactions in the MOT, multiple photon scattering, and changing lab conditions. By addressing these effects in the simulation, and improving the details in the synthetic images, the sim-to-real gap may be further closed.

## Continued learning for long-term stability

Another application of the simulation is the study of the agents under varying condition, providing insight into the long-term stability of an RL-controlled MOT. Substantial but long-term changes in the controlled system can be caused by variation of external conditions. Since retraining of the agent from scratch is time consuming, we studied how continued training during the deployment leads to continuous adaptations to various changes in the MOT. For this we divide the experiments into an initial training phase, and a subsequent so-called deployment phase. Indeed, observables (such as fluorescence images), as well as rewards can be recorded and used for further training during deployment of the operating agent. For these experiments we have used our computer simulation, which allows to operate the agent with well controlled changes to the simulated MOT. In particular, we have trained an agent (with $R \propto N/T$) for 30 k episodes, followed by subsequent episodes of the deployment phase with an additional change of the environment and deactivated exploration.

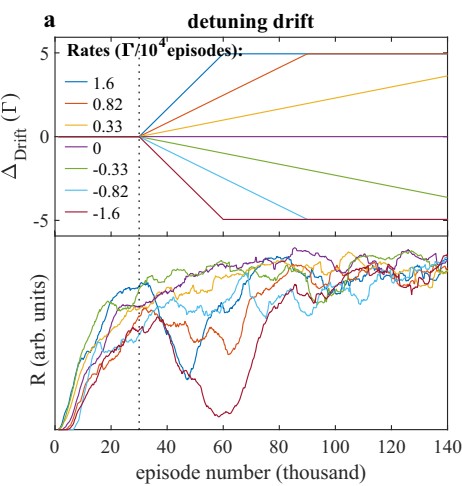
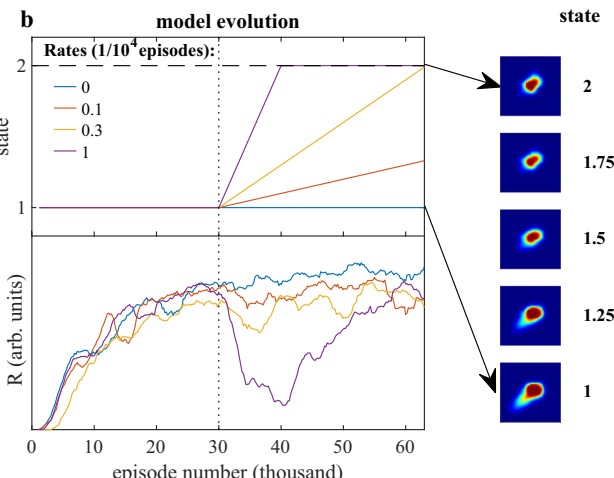

**Fig. 5 | Continued learning. a** Response of the system to a drift of the detuning, after an initial training phase (upper panel) and corresponding rewards (lower panel) as a function of the training episodes. **b** Same as in **a**, but continuously changing the fluorescence image generating model, after the initial training phase. Samples of fluorescence images generated at different stages of the model evolution are shown on the right. The dotted vertical lines indicate the start of the deployment phase at episode no. 30000, at which exploration is set to zero, but learning continues. All reward curves were smoothed using a moving average filter (size = 32) for better readability.

In the first set of simulated experiments we have introduced a drift of the detuning over the range of ±5Γ at different rates, as shown in the upper panel of Fig. 5a. This range is well beyond the random perturbation offsets $\Delta_{Offset}$ that were added during regular training. The drift stopped when ±5Γ were reached. As one would expect, for large rates of change the rewards plotted in the bottom panel started to decrease, as the absolute detuning offset quickly exceeds the range of offsets encountered during the training phase. However, as long as the drift rate was sufficiently small (below $0.33\Gamma/10^4$ episodes) the agent was capable of continuously adapting to the drifting detuning. In this case, the policy remained close to the optimal, even for large offsets reached at the end of the experiments. We terminated the drift at ±5Γ to show that continued training eventually recovered the optimal policy even when the initial drift rates were too high for continuous adaptation.

To demonstrate that our RL approach is not only capable to react to drifts in the control parameter but also to more profound changes of the environment that affect the appearance of the MOT fluorescence we have conducted a second set of simulated experiments. In real-world experiments, the appearance of the fluorescing atoms may change over time due to changing external factors such as laser intensity, polarization, or magnetic fields. Here, we used an extended generator of fluorescence images allowing to change the shape of the fluorescence as a function of an additional parameter, as depicted on the right side in Fig. 5b (see "Methods"). The actual simulated experiments were conducted analogously to the drift experiments, except that this time the image generating model was evolving over the course of the deployment phase. The capability of the agent to adapt to the changing shape of the MOT is reflected in the rewards displayed in the bottom panel of Fig. 5b. We observe that the policy remained close to the optimum, when the evolution from the initial to the final state of the model occurred over more than 30k episodes. For faster rates of evolution, the agent was "confused" by the unusual shape of the fluorescence, however, the continued training during the deployment phase recovered the optimal policy after the evolution halted. These results demonstrate that RL by design provides long-term stability for cold atoms experiments in a noisy and varying environment. Such a scenario is encountered in RL literature in the context of *Continual* or *Lifelong RL*, the goal of which is to enable the agent to adapt to new tasks or environments over time[41].

## Discussion

We have successfully shown that RL is a natural fit when the control of a cold atom experiment is formulated as a decision making problem, offering inherent robustness with respect to drifts and fluctuations at different time scales.

For this we have implemented a well studied algorithm (DDPG), running on a desktop computer and allowing for observation based and task driven real-time actions on the system. Our pioneering work will enable many labs around the world to adapt this approach to their specific needs and extend their research. In detail we have shown, that an RL agent for MOT control is able to find meaningful control strategies during a training phase and is capable to react in real-time to unexpected events, not seen during the training. Indeed, ultracold atom set-ups require stable laser frequency and intensity, protection from stray magnetic fields and precise lab temperature control[42], which poses a major challenge for the deployment of cold atom-based devices outside of lab conditions. However, the real-time feedback to the agent through fluorescence images and continued learning during deployment allows for operation under unstable conditions and constitute a central distinction with respect to other machine learning-optimized cold atom experiments. Providing additional observations of the environment such as real-time values of the laser intensity and lab temperature has the potential to further improve the performance of such a system. In this work we have facilitated the convergence of the training to comprehensible control strategies by limiting the number of parameters and their range. In future work, increasing the range, giving control to additional parameters (e.g., laser intensities and polarizations) and reducing the duration of the time steps could lead to discovery of new and possibly counter-intuitive operation modes of a MOT[14]. At the same time, allowing the agent to control additional parameters would enable the operation of MOTs for non-alkali species and molecules that require multiple lasers and elaborate control sequences. The policies discovered by RL agents would have the potential to shed new light into the physics of the light-matter interaction, especially when working with such complex systems.

In addition, we have demonstrated a paradigm change in cold atom physics by using reward engineering to focus on the goal of a specific task, instead of manually crafting control sequences. As an application of this principle, we have trained an agent that is capable of producing samples with user defined number of atoms. Conceivably, position dependent reward functions could be used to realize complex

spatio-temporal control of ultracold samples, reminiscent to RL control of a plasma in a tokamak fusion generator[29]. The promising results of transferring an agent trained on a simple simulation to the real experiment opens multiple opportunities for further research: the physical model used for the simulation can be improved by incorporating light-induced losses, with the necessary parameters readily acquired on the experiment. Domain randomization in the form of manipulation of synthetic fluorescence images could further improve the sim-to-real transfer efficiency.

With the MOT being the workhorse in the field of ultracold atoms, it presents a new, exciting platform for the development and testing of novel RL algorithms. Indeed, RL algorithms such as soft actor-critic (SAC)[43] or proximal policy optimization (PPO)[44] can be readily implemented and tested in our simulation and brought to the experiment. To foster progress in this new and exciting field, we made the code for training and evaluating an agent based on a simulated MOT publicly available (https://github.com/MPI-IS/RL-coach-for-MOT/releases/tag/1.0). Finally, our work constitutes the first step to the production of degenerate quantum gases using RL control, provided that suitable real-time observables (e.g., ion signals[45] or non destructive imaging methods[17,46,47]) can be implemented.

Note: While this manuscript was under review, a related study was published[48], demonstrating the control of a quantum gas experiment using a single-step RL agent.

## Methods

### MOT operation and implementation

MOT operation requires a quadrupole magnetic field and in the most common configuration a pair of counter-propagating laser beams (cooling beams) in all three spatial directions. Together, this makes for a position and velocity dependant light force (radiation pressure), which cools and traps neutral atoms from the gaseous phase at the center of the magnetic quadrupole field where the beams are overlapping.

Our MOT system is based on $^{87}$Rb atoms and is described in the following. The quadrupole field is generated by a pair of coils in Anti-Helmholtz configuration, providing a field gradient of $B' = 15\,\text{Gcm}^{-1}$ at the quadrupole center, along the coil symmetry axis for about 1 A current in the coils. The coil system is placed inside a UHV science chamber at a pressure of $10^{-11}$ mbar and operated via a voltage controlled current source. The cooling beams with waists ($1/e^2$-beam radius) of 12 mm and powers of about 15 mW are circularly polarized and overlap at the center of the magnetic quadrupole between the coils. The MOT is loaded from a beam of precooled Rb atoms emitted from a two-dimensional (2D)-MOT-source. The latter consists of field coils and thermal rubidium emitter sources (dispensers), providing a rubidium partial pressure of $>10^{-9}$ mbar in the corresponding 2D-MOT chamber. It is attached to the science chamber via a differential pumping stage with a direct line of sight between the centers of the two MOT systems. The 2D-MOT is loaded with rubidium atoms from the background pressure and creates a transversally cooled beam of cold atoms guided to the MOT in the science chamber.

As cooling and trapping relies on (position and velocity dependant) radiation pressure due to light scattering, MOT operation requires a closed cooling cycle transition for the atoms. That is a state pair, which the atoms cannot leave during absorption or spontaneous emission. In $^{87}$Rb such a transition is found in the D2 manifold between the $5S_{1/2}$ ground state and the $5P_{3/2}$ excited state, near 780 nm. Both states are split in two respectively four hyperfine manifolds with total angular momentum $F = 1, 2$ and $F' = 0, 1, 2, 3$. We use the $5S_{1/2}, F = 2 \rightarrow 5P_{3/2}, F' = 3$ cooling transition, which is considered to be closed. For MOT operation, the cooling light is red detuned to this transition. Atoms can leave the otherwise closed cooling cycle only via off resonant excitation to the $5P_{3/2}, F' = 2$ state and adjacent decay to the $F = 1$ ground state. A repump laser resonant to the $5S_{1/2}, F = 1 \rightarrow 5P_{3/2}, F' = 2$

transition and overlapped with the cooling light is used to bring back the atoms to the cooling cycle.

The repump beams are superimposed onto the cooling beams. Each axis has a power of ~3 mW, is circularly polarized, and has a beam waist equal to the cooling beams. The 2D-MOT is operated with separate cooling and repump beams at the same frequencies.

### Control implementation

The control of the experimental apparatus is divided between the real-time system Jäger Adwin Pro II and a desktop PC dedicated for the RL agent. While the Adwin system is responsible for the time-of-flight imaging operations, which require exact timing, the desktop PC controls the selected experimental parameters ($\Delta$ and $B'$) via a digital-to-analog converter (DAC) card (AdLink DAQ-2502). The analog output line of the DAC card for the detuning $\Delta$ controls directly the offset lock used for frequency stabilization of the laser. $B'$ is set via further analog output lines that control the current in the MOT-coils.

An episode is initiated from the PC via enabling the magnetic field gradient, effectively turning the MOT on. An additional output line from the DAC card to the Adwin system is used to trigger the time-of-flight imaging sequence at the end of an episode. At the conclusion of the imaging sequence, the Adwin system returns into the idle state, ready to start loading atoms.

The fluorescence camera (UEye UI-3060CP) is connected to the PC and is constantly recording images and transferring them into a reserved memory register at a rate of 150 fps. During an episode, in each time step the latest available image is read out from the memory and processed to be passed as part of the observation to the agent.

### Fluorescence imaging

As atoms in the MOT scatter laser light during the cooling and trapping process, the scattered light can be observed as fluorescence. This phenomenon makes the atomic cloud in the MOT observable, with the spatial distribution of atoms being reflected in the appearance of the fluorescence. Typically, the extension of the fluorescing cloud is on the order of millimeters. The rate at which fluorescence is emitted depends on the laser detuning and is modulated by the magnetic field gradient. This results in a highly non-linear mapping of the 3-dimensional atomic density distribution onto the imaging plane. The images then correspond to a projection along the imaging axis and show the fluorescence intensity in the direction of the camera. Different processes such as radiation trapping or imbalanced radiation pressure can lead to a variety of non-trivial shapes of the atomic cloud, including ring structures[49].

The fluorescence images have a dimension of $50 \times 50$ pixels with 8-bit resolution. The exposure settings are chosen such that even weak signals can be detected, which is important for the initial phase of the learning. This leads to fast saturation when loading at maximum rate.

We have conducted tests in the simulated environment providing the agent only with integrated fluorescence signals instead of the full image as observable. The training in such a scenario did not converge to the optimal policy, in particular, the agent was not able to adapt to any perturbation.

### Reference episodes

The reference episodes, in which the control sequence was manually set to the intuitive optimal policy, i.e., loading at optimal detuning and ramping the detuning to the maximal value in the last time step, allowed us to normalize the evolution of the performance (i.e., reward $R$), correcting for possible long term drifts of the experimental apparatus. In practice it means that in each time step of a reference episode the actions of the agent were overwritten with predefined values while keeping the timing constraints of the agent, i.e., the 60 ms step size. The reference episodes were included during training (see below) and when evaluating a trained agent.

## RL algorithm

For the realization of an RL-controlled MOT we have taken recourse to a modular framework known as *coach*[32], containing implementations of a variety of RL algorithms in Python. In this work, we have focused on the DDPG algorithm, an actor-critic algorithm, that is able to learn directly from raw images and uses experience replay for training[35]. Experience replay involves storing experiences, which contain observations and rewards for a single time step, in a memory buffer as an agent interacts with the environment. Instead of using each new experience immediately for learning, the training algorithm samples a batch of experiences from the replay memory after each training episode and uses these samples to update the agent's policy. We have found that almost all parameters are close to the optimum at their default values (as defined in *coach*). The parameters used for the simulation can be found in our preset published as a part of the code (https://github.com/MPI-IS/RL-coach-for-MOT/releases/tag/1.0).

Inspired by the performance of the Deep Q-Network[19] we have adapted the stacking of the 4 most recent fluorescence images as input to the agent. This allows the agent to compute the rate of change (and higher derivatives) of the fluorescence signal, which can be used to determine the loading rate of the MOT. All values passed as observables are normalized to the range [0,1].

## Scheduling

The learning starts with a heat-up phase of 400 episodes during which the control is fully determined by a Brownian motion-like exploration. During the subsequent training phase, every 200 training episodes, a set of 10 reference and 10 evaluation episodes with random perturbation offsets are recorded. The performance of the agent is evaluated by disabling the exploration during the evaluation episodes. During the training episodes, unlike in the original DDPG implementation, we compute a new action in every time step (i.e., no *action repeat*).

## MOT simulation

We have implemented a phenomenological, data-driven computer simulation for the MOT in order to better understand the behavior of the agent in such a novel environment, as well as to investigate sim-to-real transfer.

For our simulation we assume that the number of trapped atoms increases by a certain number $dN$ in each time step, where $dN$ only depends on the applied control parameter $\Delta_{n_t}$ at time step $n_t$.

$$N(n_t) = \sum_{i=1}^{n_t} dN(\Delta_i) \qquad (2)$$

This means that we neglect loss processes that are known for MOTs and typically lead to saturation of the accumulated number of atoms. Furthermore, we assume that at each time step the temperature of the trapped cloud is dictated by the cooling laser light and thus only depends on the value of the detuning.

$$T(n_t) = T\left(\Delta_{n_t}\right) \qquad (3)$$

This assumption is valid as long as the atomic cloud stays dilute and transparent to the light. Thus, we neglect density dependent multiple scattering. In order to obtain the most accurate simulation, we experimentally determine the loading rate $dN$ as well as the temperature $T$ for a discrete number of detuning values and create look-up-tables (LUT), which are used for interpolations during the simulation (cf. Fig. 4a). The loading rate as function of the detuning is measured by setting the desired value $\Delta$ for a duration of 1 s and measuring the final number of atoms via absorption imaging. The temperature LUT is recorded by loading atoms at the optimal loading rate and subsequently rapidly ramping the detuning to a given value before performing absorption imaging. The fluorescence images that are captured right before the measurement of $dN$ and $T$ via absorption imaging are stored in a database. Each measurement is repeated 5 times.

The full dataset contains on the order of 5000 samples. The environment based on the simulation computes the intermediate values of $dN$ using a linear interpolation. The temperature $T$ is obtained by interpolating between the measured values using an exponential function. During the training on the simulation, noise is introduced to the loading rate $dN$ in order to simulate the physical fluctuation of the experimental conditions.

$$dN \rightarrow dN \cdot \mathcal{N}\left(\mu, \sigma^2\right) \qquad (4)$$

with $\mathcal{N}$ being the normal distribution at mean value $\mu = 1$ and standard deviation $\sigma = 0.1$.

We trained a convolutional neural network (CNN) to generate realistically looking fluorescence images for the simulation on the same experimental dataset. We have implemented the decoder-part of an autoencoder network[50,51] as the generator and performed supervised training with $N$ and $\Delta$ as inputs and the fluorescence image as output. The network consists of a fully connected layer, followed by 5 convolutional layers with rectified linear unit as activation function and upsampling layers in between. Prior to the training we set fluorescence images to 0 for samples with $N = 0$, as well as remove outliers from the dataset. A small, randomly chosen subset of the dataset was held out during training and used for testing the generator. In Fig. 6, we compare the recorded images to the output of the generator for several parameter sets. The overall appearance of the generated images shows high similarity to the real images, with some fine grained details not fully captured by the CNN.

The extended model used in the long-term stability simulations was trained on a dataset consisting of two subsets of fluorescence images, that were recorded on two dates that lay several weeks apart

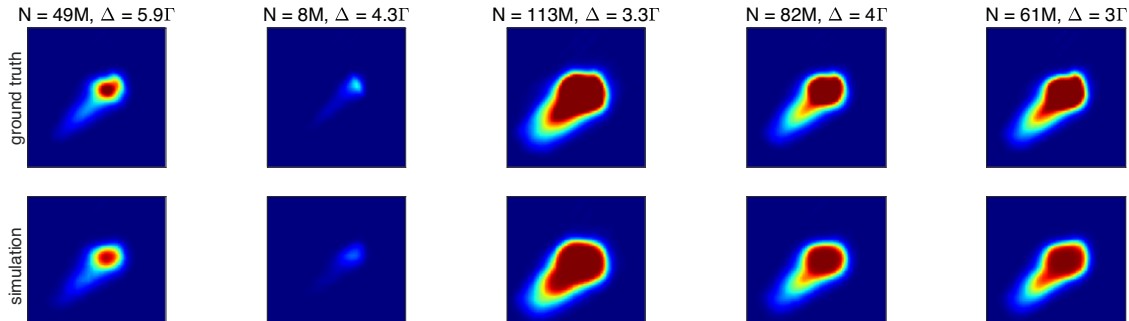

**Fig. 6 | Realistic synthetic MOT images.** Experimental fluorescence images from the test dataset (upper line) and corresponding synthetic images (lower line) for different parameter sets ($N$, $\Delta$).

and therefore showed noticeably different shapes of fluorescence for the same $N$ and $\Delta$. For the training, each fluorescence image was labeled not only with $N$ and $\Delta$ but also with the parameter (1 or 2) indicating the subset of images it belonged to. This way, the resulting CNN was able to reproduce images from both subsets, and more importantly, was interpolating between the two, when an intermediate parameter value was provided.

## Data availability

The data that support the findings of this study are fully available on request from the authors.

## Code availability

Our code for training and evaluating an RL agent based on a simulated MOT can be found under (https://github.com/MPI-IS/RL-coach-for-MOT/releases/tag/1.0).

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

## Acknowledgements
The authors acknowledge fruitful discussions with G. Martius. The authors thank J. Tiggemann and F. Holler for their valuable help as student assistants. A.G., J.F., and M.R. acknowledge funding from the German Research Foundation (DFG) through the Research Unit FOR 5413/1, Grant No. 465199066.

## Author contributions
A.G., J.F., and V.V.V. planned and supervised the project. M.R. carried out the experiments. V.V.V. carried out the simulations. M.R., A.G., and V.V.V. analyzed the data. M.R., A.G., and V.V.V. wrote the manuscript. All authors discussed the results and contributed to the manuscript.

## Funding

## Competing interests
The authors declare no competing interests.
