## [Peer Review File · Nature Communications]

Reinforcement Learning in Cold Atom ExperimentsREVIEWER COMMENTS

Reviewer #1 (Remarks to the Author):

In the manuscript "Reinforcement Learning in Ultracold Atom Experiments", Reinschmidt et al. describe an implementation of machine learning optimization of the preparation of a cold atom cloud. Machine learning optimization methods have gathered a lot of interest as of late in a wide variety of fields, including cold atoms, where the complexity of the experimental procedures inherently limits the realistically accessible parameter space for human exploration. They have also shown the potential to uncover non-intuitive solutions (i.e. non-adiabatic). The present manuscript goes one step further.

Specifically, reinforcement learning (RL) is used, and a significant novelty is live observation during the MOT loading sequence based on fluorescence imaging. One or two experimental parameters are optimized over 25 time steps per sequence/episode. On its own, the agent learns to optimize the MOT (in terms of temperature and atom number) with sequences similar to those a physicist would converge to. The results are absolutely convincing, with a thorough analysis of learning process. It will be a nice additional tool for the community, especially with the "real-time" optimization from fluorescence imaging that can react to changes in the environment. However in this instance, very little new physics is extracted in the sense of unexpected high performance strategies, detailed study of the stability of optimal strategies, of their physical meaning, all of which would have significant additional impact.

The manuscript reads very well, but it would benefit from additional context with respect to previous implementation of ML in cold atom setups, as well as some details as to why some of the static experimental parameters were chosen as such (details below). Thus, in the current state, I'm unsure of the impact of these results.

Detailed comments

1- I find myself confused by the underlying opposition between reinforcement learning (without training input) and previous implementations (Refs 10-15), in the first paragraph. In those previous studies, supervised learning appears to have been used, yet with the training eventually also generated by an algorithm rather than fed by the user. Thus, viewed from the outside as kind of black boxes, the two appear very similar, albeit for the "live" optimization.

2- The exact role of fluorescence monitoring is also unclear. Does it allow truly live optimization? It appears as a nested optimization with an a priori undefined reward from the images that is tuned to maximize the final reward. Can you comment? How does this relate to RL?

3- One question arising from the optimization from fluorescence imaging, likely beyond the scope of this paper, is how information about atom number and temperature reflect inside the fluorescence images, which are highly distorted by saturation and off-resonance scattering at non-zero B-Field.

4- The choice of parameters is odd at times and likely limiting the final performance, specifically with the repumper power being very strong, i.e. the MOT is "bright", where a kind of "dark MOT" typically leads to larger, colder MOTs, by reducing the reabsorption probability. I understand that choosing few parameters, and these ones in particular, one ends up in a well controlled textbook-like situation. But this also strongly limits the accessible new physics.

5- It would be nice to see how stable/broadband the B' ramps from Figure 2d are. Especially what is really the improvement from varying it that way? Essentially, is it minor or really significant? And why?

6- The reward function targeting a specific atom number is indeed nice. It is highlighted as new, yet a

similar optimization was performed by Barker et al. (Ref 13).

7- Some figures could be improved for clarity: Figures 2a-b-c could use a description inside the figures, Figure 2d would benefit from a visual cue of what curve corresponds to what vertical axis, and Figures 2e and 4b-c could use legends (even though the caption is clear).

8- I noted two typos lines 299 "could be expected" and 302 "as a consequence".

Reviewer #2 (Remarks to the Author):

Referee report

This research paper presents a novel approach to optimize the preparation and control of atomic clouds in cold atom traps using reinforcement learning. The authors introduce a semi-continuous, in-sequence non-destructive state measurement using the MOT fluorescence level to allow for the learning of real time adaptation strategies for shot to shot experimental parameter drifts. They optimize both one and two parameter systems and demonstrate that numerical simulations can be transferred to experimental settings.

Overall, the paper introduces a very interesting approach and results very relevant for the field. I was certainly very excited to read about and learn from the methods and results. However, as someone who has been dabbling in some related areas, I must say that it does not seem to me that the authors manage to communicate clearly to a broad readership what the main contribution of the paper is and the level of transferable insights and to both novices and experts in the field, I believe that the impact of the paper could be improved greatly by increasing the depth of the methodological discussions by comparing and contrasting more clearly with established techniques. I will give a few examples from my own personal perspective below in the detailed comments. To me, the answer to these questions/comments should determine whether the results belong in the current journal or in a more specialized one.

Detailed comments:

- One of my concerns about the manuscript can be illustrated with the proposed title: it is a very generic title given the rather mature field. Do the authors claim that their main novelty is the application of reinforcement learning rather than deep ML previously applied in the field? If so, then the suitability of the manuscript to the journal should be judged by the amount of papers employing RL techniques broadly speaking in quantum physics and whether the extension to cold atoms and the methodological extension from other ML to RL is significantly interesting to a broad readership. To me, it seems from the introduction that it is rather the in-sequence (real-time) training enabled by the continuous fluorescence measure capture and feedback that enables a different approach compared to previous papers in the field. In that case, it is not clearly argued why the chosen method is necessary or optimal compared to other possible methods (see next comment). Also, given the prior work on BECs eg ref 12, it seems quite a stretch to use the title ultracold for the MOT experiments presented here. The mentions of cold atom physics in the abstract seems more appropriate.

- Competing approaches background: if it is indeed the intention of the authors that the main conceptual and experimental innovation is the addition of the continuous measurement and training, then it would seem appropriate to highlight clearly the two main features of the approach: that drifts exist and that if these were known, it would allow for the application of state specific optimal strategies to be employed. In this formulation, it would clearly illustrate the competing approaches that are not at all discussed in manuscript:

- Competing approaches 1: first, I think it would greatly assist the reader if the method was compared to active parameter estimation and feedback approaches. Personally, I find that the discussion of the single parameter (frequency) case in the paper is very similar to a conventional feedback loop locking scheme in which one could imagine measuring the fluorescence signal and then transforming it appropriately and sending it back to the laser frequency. Even a complex transformation can be done now a days routinely with FPGAs. The main question is: what does the fluorescence signal say about

the "hidden" parameters. I think the main contribution of this part of the manuscript is to outline a no-human-in-the-loop determination of this non-linear transfer function but given the centrality of the real time measurement I believe that a much more thorough review of previous approaches examining the signal contained in the fluorescence signal. In particular because it has over the past two decades, I believe, been the desire of every MOT-graduate student to simply hook the fluorescence signal to an active feedback stabilizer and deliver the more stable atom number output that the authors discuss in the second half of the paper.

- Competing approaches 2: when a significant part of the results is exactly the application to this atom number stabilization using continuous measurement, I find it remarkable that the authors do not compare and contrast their approach to the broader approach of applying QND measurements and feedback. Here in particular it would be relevant to refer to some of the state of the art work eg correcting for a large number of experimental drifts to achieve stabilization down to shot noise level of BEC numbers, <https://journals.aps.org/prl/abstract/10.1103/PhysRevLett.117.073604>

- Competing approaches 3: once the issue of determination of potential drift parameters is settled, it would make sense to more clearly discuss the background for the second challenge, the determination of the optimal strategies given these parameters. Here it would be important to give the readers a basic understanding of the relation to basic optimization concepts like gradient based optimization (primarily for simulations) and gradient free approaches such as the very simple remote dressed CRAB approach explored in <https://www.pnas.org/doi/10.1073/pnas.1716869115> for the optimization of ultra cold atoms and then moving on to the ML based approaches briefly mentioned in the introduction. Providing strengths and weaknesses (eg convergence properties, ie number of training/optimization runs necessary) for these approaches will greatly help the reader understand the significance of the current work. Therefore, I also believe that more detail should be added around the selected references relevant for the cold/ultracold experiments such as ref 11 and 12.

- Another reason for including a brief discussion about optimization is that it introduces the cost function and thus nuances the statement in the abstract and paper that one of the main novelties is the manipulation of the reward function. I agree that it is a strength of the RL paradigm to engineer the reward function but it is not true that, given the right insights, related results could not be obtained using an optimization technique. As mentioned above, the transfer function may however be complicated and this is where the current approach shines.

- Competing approaches 4: one of the main advantages of the ML/RL paradigm compared to other approaches is exactly the "discovery" of novel insights. Compared to eg the discussion in ref 12 about "discovering" redundant parameters I find the physical insights discussions of the current paper could be improved. For instance, it is my impression that in the 1D case of detuning variations, the algorithm "discovers" the importance of maximizing the loading rate and uses a pair of fluorescence measurements to estimate the current rate and then calculate the detuning jump necessary to obtain optimal loading rate. Is this true or is it more complicated than this?

- Misc comments:

- o Page 1 line 44: In providing the list of previous RL papers, since the following paper is to my knowledge the first paper employing the deep RL technique AlphaZero to quantum physics earlier than all of the referenced RL in physics papers (refs 23-227), it is somewhat odd not to cite it: <https://www.nature.com/articles/s41534-019-0241-0>

- o Page 4 line 249: not clear how the variations in control policy are realized? is it by running a number of training events each w $\sim 20k$ episodes, or by evaluating the same converged agent/policy on a number of experimental realizations? i assume the former, but it is not clearly described how many times this was run. isnt the lack of variation a sign of the simplicity of the problem rather than proof that the "control policy is remarkably stable"?

- o Page 4 line 250: again it is not clear how the error bars in fig 2e are obtained.

- o Page 5 line 314: not so surprising, given there is more temporal freedom. more interesting is why this gain only happens for negative detunings. what is the physical insight gained from this?

- o Page 6, line 362: again, the comparison to other active control strategies has not been made clear. in particular, it is not clear how this method would compare in performance to either an active lock to the loading rate (adjusted for detuning dependent fluorescence rate) or even simpler a tabulated loading rate vs detuning single step look up. such implementations are somewhat complex to implement using

analog control electronics but easy to implement on eg an FPGA.

these are comments/ideas from my personal perspective to highlight more clearly the novelty of the pursued approach and i hope they will be helpful in the resubmission process.

best regards
Jacob Sherson

Reviewer #3 (Remarks to the Author):

The authors use reinforcement learning to control a MOT. They show how the system can be trained both from real data and in-silico and demonstrate a robust performance of the hardware + software loop. The main difference from previous implementations of similar ideas is that there is not pre-determined experimental sequence, instead the network adaptively controls the experiment based on intermediate non-destructive measurements.

The work is nicely done and I appreciate that the authors have made available both the data, simulations, and code for other interested parties to reproduce their results. I generally find that the work presented here is in line with the recent trend of using neural networks for tasks that do not necessarily benefit from them. Although the NN part of the work is a tour-de-force the physics part remains largely unconvincing. The authors control one or two parameters of the system only and cannot reproduce/discover standard techniques like cMOT because of the slow fps.

Major comments.

- Reinforcement learning algorithms are notoriously known for the amount of samples they need for training. That might be fine for a MOT sequence but I suspect it will start becoming a problem in a full BEC cycle. Already the training of a MOT takes up to 2 days. The authors also never discuss how long for that training remains valid for. In the end of the day all this learning is contingent to the experimental stability and reproducibility. How stable do conditions in the lab need to be for the NN to keep working? How long does it actually keep working for before it needs to be retrained.

- I am assuming that the authors are after doing BEC physics ultimately (please forgive me if this assumption does not hold) however, experientially the MOT atom number is only weakly correlated with the BEC atom number. The later can be and has been stabilised to the point of being limited by Poissonian fluctuations in Phys. Rev. Lett. 122, 163601 and related work from the same group using a simple feedback loop with any NN.

- The authors only control one or two parameters. Even in a MOT these are too few. There is power and detuning of cooling and repumper lasers, polarisations, control of beam imbalance, etc. I'm guessing adding more parameters would make the training time prohibitively long. It might be more interesting to control the shape of the MOT however that the atom number in order to make subsequent stages more efficient. In any case, due to the slow fps, some standard techniques like cMOT will always be out-of-reach for the NN. Moreover, the NN cannot detect any fluctuations faster than 60Hz if they are synchronous with its sampling rate.

- I find the presentation of the NN as completely sequence-agnostic somewhat misleading. It's only due to the MOT being a stable object with infinite lifetime and a human having made a number of decisions about all the other parameters that make it possible and about the limits of the parameters the NN explores. The distinction of those decisions not involving time is a pretty fine one.

- The authors miss a key reference, A J Barker et al 2020 Mach. Learn.: Sci. Technol. 1 015007, where various NN techniques were explored for BEC optimisation. This somewhat diminishes the novelty of the work.

- During the training phase, the authors put artificial offsets to make the NN more robust. Changes like these are synchronous and are passed to the next timestep. If the field was to have a large spike and return to the same value the algorithm would be tricked. Why not change some real environmental parameters to emulate actual hardware variation?

Minor comments.

- Episodes seems to be coach-specific jargon. Why not epochs?

- It wasn't clear to me in what sense are the fluorescence images not part of the cost function. Why are they being taken every 60ms then?

- Is the processing time part of the 60ms cycle or is there an additional dead time as well?

- Panels in figures 2 and 3 are referenced out of order which makes for a confusing read.

- I would be interested to see MOT loading curves for the atom-number-fixing experiments. Has the NN understood to drive the MOT into saturation and hence reduce the number fluctuations or is stopping the loading on a positive slope?

- 60ms sampling is not the same as real-time feedback and I believe the language should be changed in the manuscript. It might be faster than typical experimental cycles in atomic physics but at the same time it's too slow for discovering sharp-change sequences.

- It's odd that capturing the total fluorescence didn't work well. Why? What feature of the fluorescence images is used in the end? Also I thought they weren't used in the reward.

Reply to the Reviewers

Re: Manuscript ID NCOMMS-23-18686

“Reinforcement Learning in Ultracold Atom Experiments”

Malte Reinschmidt , József Fortágh , Andreas Günther, and Valentin Volchkov
Nature Communications

Reviewer COMMENTS

Reviewer #1

In the manuscript "Reinforcement Learning in Ultracold Atom Experiments", Reinschmidt et al. describe an implementation of machine learning optimization of the preparation of a cold atom cloud. Machine learning optimization methods have gathered a lot of interest as of late in a wide variety of fields, including cold atoms, where the complexity of the experimental procedures inherently limits the realistically accessible parameter space for human exploration. They have also shown the potential to uncover non-intuitive solutions (i.e. non-adiabatic). The present manuscript goes one step further.

Specifically, reinforcement learning (RL) is used, and a significant novelty is live observation during the MOT loading sequence based on fluorescence imaging. One or two experimental parameters are optimized over 25 time steps per sequence/episode. On its own, the agent learns to optimize the MOT (in terms of temperature and atom number) with sequences similar to those a physicist would converge to. The results are absolutely convincing, with a thorough analysis of learning process. It will be a nice additional tool for the community, especially with the "real-time" optimization from fluorescence imaging that can react to changes in the environment. However in this instance, very little new physics is extracted in the sense of unexpected high performance strategies, detailed study of the stability of optimal strategies, of their physical meaning, all of which would have significant additional impact.

The manuscript reads very well, but it would benefit from additional context with respect to previous implementation of ML in cold atom setups, as well as some details as to why some of the static experimental parameters were chosen as such (details below). Thus, in the current state, I'm unsure of the impact of these results.

Our response #1.0

We thank the Reviewer #1 for his largely positive feedback on our manuscript and are grateful for his comments and suggestions. They helped us to clear ambiguities and to sharpen the manuscript's language.

With our detailed answer to the comments and the corresponding changes to the manuscript, we hope to convince the Reviewer, that RL based control of cold atoms setups is conceptually different from previous implementations of ML in this field, thus having a strong potential for a significant impact on future developments.

Detailed comments

Reviewer #1, comment #1

I find myself confused by the underlying opposition between reinforcement learning (without training input) and previous implementations (Refs 10-15), in the first paragraph. In those previous studies, supervised learning appears to have been used, yet with the training eventually also generated by an algorithm rather than fed by the user. Thus, viewed from the outside as kind of black boxes, the two appear very similar, albeit for the "live" optimization.

Our response #1.1

We agree that the previous implementation of machine learning methods (Refs 10-15) used training samples that were generated during the training process and therefore can be seen as examples of self-supervised learning, similar as in reinforcement learning. We thank Reviewer #1 for pointing it out and have extended the discussion of previous studies in the main text in order to emphasize the fundamental differences with respect to reinforcement learning (lines 29-48, 56-63, 76-79). The "live" property is one of the main differences and reflects that an RL agent is conceptually a feedback controller that acts in each time step according to its policy and based upon the observables, while the other approaches optimize the entire control sequence at once and are thus unable to react to changes in the environment.

Reviewer #1, comment #2

The exact role of fluorescence monitoring is also unclear. Does it allow truly live optimization? It appears

as a nested optimization with an a priori undefined reward from the images that is tuned to maximize the final reward. Can you comment? How does this relate to RL?

Our response #1.2

The role of fluorescence images is crucial to the working principle of RL as the images constitute the main input to the controller (agent), enabling live-feedback to the MOT, **and** are used in the learning of the policy. Thanks to the comment of Reviewer #1, we realized that we neglected the latter aspect, which is slightly technical but nevertheless important. We now explicitly state this relation in the text in lines 92-95,173-175,188-191. We have also added a drawing to Fig. 1 in order to better illustrate how fluorescence images are used in the control process. Furthermore, we added details regarding how the fluorescence images are used in the process of policy optimization in the Methods section (lines 790-798). In brief, the optimization takes place in between the training episodes and consists of updating the policy using batches of observation-reward pairs sampled from all previously experienced time steps. In this sense, the optimization of the policy is not truly live, since the most recent experienced time steps have an impact only after certain number of training steps. This turns out to be more stable, than directly learning from the most recent time steps.

Regarding the "undefined reward", we assume that Reviewer #1 is referring to rewards that are assigned to observations of time steps other than the terminal step of an episode. From a technical point of view, the reward value of 0 is attributed after each time step but the last. In RL the apparent problem of "undefined reward" for time steps that are not terminal is handled via a so-called *value function*, which represents for a given time step the expected final reward the agent can obtain. However, this subject is very extensive and goes beyond the scope of our work, which is why we decided not to address it in our manuscript.

Reviewer #1, comment #3

One question arising from the optimization from fluorescence imaging, likely beyond the scope of this paper, is how information about atom number and temperature reflect inside the fluorescence images, which are highly distorted by saturation and off-resonance scattering at non-zero B-Field.

Our response #1.3

As the Reviewer #1 points out, the image of the fluorescing MOT depends not only on the number of atoms but also on B-field configuration, the detuning, and many other parameters such as beam profiles, polarizations, camera saturation and effects like radiation trapping. Since the agent is trained end-to-end with the goal to maximize the reward, a priori the agent does not operate with concepts such as the number of atoms or the temperature (or any other physical property of the system). The agent tries to figure out the features of the image that give rise to the strongest gradients of the value-function (derived from the reward).

Reviewer #1, comment #4

The choice of parameters is odd at times and likely limiting the final performance, specifically with the repumper power being very strong, i.e. the MOT is "bright", where a kind of "dark MOT" typically leads to larger, colder MOTs, by reducing the reabsorption probability. I understand that choosing few parameters, and these ones in particular, one ends up in a well controlled textbook-like situation. But this also strongly limits the accessible new physics.

Our response #1.4

For the first proof-of-principle demonstration of an RL controlled MOT, we have deliberately chosen only few control parameters and otherwise use typical values for the remaining parameters. This includes the intensity in the individual cooling and repumper beams, which amount to approximately $2I_{sat}$ and $0.5I_{sat}$, respectively. We have corrected a previously erroneous specification of the repumper power in the Methods section (line 711).

Based on our experience from the presented work we are currently working on a dedicated experimental setup where parameters like repumper detuning, power etc. are also controlled by the agent, which should increase the potential for (re-)discovery of non-intuitive cooling strategies. The exploration of the intriguing potential of such a setup will be the focus of forthcoming publications.

Reviewer #1, comment #5

It would be nice to see how stable/broadband the B' ramps from Figure 2d are. Especially what is really the improvement from varying it that way? Essentially, is it minor or really significant? And why?

Our response #1.5

The stability of the B' ramps is shown in Fig. 2d (now e) for a single offset value, with the shaded area corresponding to the standard deviation of multiple repetitions. The variation in the B' ramps reflects the weak dependence of the MOT loading on the value of the gradient in the allowed range. We attribute the increase of the gradient at the beginning of an episode to an increased sensitivity of the fluorescence to the detuning at high field gradients. We have added this interpretation in the revised manuscript (lines 351-353). We observe no significant improvement from the obtained B' ramps, however, as we point out in the revised text (lines 357-359), the main results of this subsection consists in the fact that the agent was able to recover the near-optimal detuning control sequence despite the increased action space.

Reviewer #1, comment #6

The reward function targeting a specific atom number is indeed nice. It is highlighted as new, yet a similar optimization was performed by Barker et al. (Ref 13).

Our response #1.6

We are delighted that Reviewer #1 appreciates the promise of reward function engineering presented in our work. Targeting a specific atom number is an example of it and is closely connected to the work of Barker et al. (Ref. 13). We have added a discussion about this in the revised manuscript (lines 446-451) as well as a comparison to other methods (lines 375-387, Refs. 41,42).

The main advantage of our approach is that once the training is completed, any atom number in a large interval can be produced within a given episode. This is possible because the requested number is provided to the agent along with observations at run-time. Such a mechanism is missing in Barker et al., where the specific number of atoms or temperature is "hard-coded" into the cost-function. Changing the targeted number of atoms would require Baker et al. to re-train the system with a new cost-function.

Reviewer #1, comment #7

Some figures could be improved for clarity: Figures 2a-b-c could use a description inside the figures, Figure 2d would benefit from a visual cue of what curve corresponds to what vertical axis, and Figures 2e and 4b-c could use legends (even though the caption is clear).

Our response #1.7

We thank the Reviewer #1 for these suggestions, especially regarding Figure 2d (now e). We have included description inside the Figure 2 a-d, colored the vertical axes of Figure 2d (now e), and added legends to Figure 4 b-c.

Reviewer #1, comment #8

I noted two typos lines 299 "could be expected" and 302 "as a consequence".

Our response #1.8

We thank the Reviewer and have fixed both typos.

Reviewer #2

This research paper presents a novel approach to optimize the preparation and control of atomic clouds in cold atom traps using reinforcement learning. The authors introduce a semi-continuous, in-sequence non-destructive state measurement using the MOT fluorescence level to allow for the learning of real time adaptation strategies for shot to shot experimental parameter drifts. They optimize both one and two parameter systems and demonstrate that numerical simulations can be transferred to experimental settings. Overall, the paper introduces a very interesting approach and results very relevant for the field. I was certainly very excited to read about and learn from the methods and results. However, as someone who has been dabbling in some related areas, I must say that it does not seem to me that the authors manage to communicate clearly to a broad readership what the main contribution of the paper is and the level of transferable insights and to both novices and experts in the field, I believe that the impact of the paper could be improved greatly by increasing the depth of the methodological discussions by comparing and contrasting more clearly with established techniques. I will give a few examples from my own personal perspective below in the detailed comments. To me, the answer to these questions/comments should determine whether the results belong in the current journal or in a more specialized one.

Our response #2.0

We would like to thank the Reviewer #2 for his very detailed report on our work and his positive evaluation. We are extremely thankful for his numerous constructive suggestions and comments, which largely helped us to improve the manuscript. In particular, pointing out the insufficient comparison and differentiation to other methods (as also commented by Reviewer #1), has given us the opportunity to significantly rework the manuscript and strengthen the focus of our work. In this context, we are specifically thankful, for the suggestions to deepen the methodological discussion by comparing and contrasting our method more clearly with established techniques. We happily followed these suggestions in our revised manuscript, and are convinced, that they strengthen the impact of our work.

Detailed comments:

Reviewer #2, comment #1

- *One of my concerns about the manuscript can be illustrated with the proposed title: it is a very generic title given the rather mature field. Do the authors claim that their main novelty is the application of reinforcement learning rather than deep ML previously applied in the field? If so, then the suitability of the manuscript to the journal should be judged by the amount of papers employing RL techniques broadly speaking in quantum physics and whether the extension to cold atoms and the methodological extension from other ML to RL is significantly interesting to a broad readership. To me, it seems from the introduction that it is rather the in-sequence (real-time) training enabled by the continuous fluorescence measure capture and feedback that enables a different approach compared to previous papers in the field. In that case, it is not clearly argued why the chosen method is necessary or optimal compared to other possible methods (see next comment). Also, given the prior work on BECs eg ref 12, it seems quite a stretch to use the title ultracold for the MOT experiments presented here. The mentions of cold atom physics in the abstract seems more appropriate.*

Our response #2.1

Indeed, our main claim is that application of RL in the field of cold atoms is novel and allows for adaptive real-time control of the experiments. We agree with Reviewer #2 that the term "cold" (associated with thermal gases, as is our MOT) is more appropriate for the title than "ultracold" (often used in the context of degenerate gases) and have modified the title accordingly.

Reviewer #2, comment #2

- *Competing approaches background: if it is indeed the intention of the authors that the main conceptual and experimental innovation is the addition of the continuous measurement and training, then it would seem*

appropriate to highlight clearly the two main features of the approach: that drifts exist and that if these were known, it would allow for the application of state specific optimal strategies to be employed.

Our response #2.2

Thanks to the Reviewers comment, we became aware, that we have not highlighted sufficiently the importance of experimental drifts and perturbations in the context of our work. Therefore, we have added emphasis on how our results contribute to new strategies for the handling of drifts on various timescales (lines 103-107). In this context we also have provided a new section describing how long-term drifts can be mitigated by continued reinforcement learning (lines 514-583). Finally, we have also strengthened the focus on applications such as MOT-based devices that are subject to external perturbation throughout the manuscript (lines 23-26, 598-603).

Reviewer #2, comment #3

In this formulation, it would clearly illustrate the competing approaches that are not at all discussed in manuscript:

- Competing approaches 1: first, I think it would greatly assist the reader if the method was compared to active parameter estimation and feedback approaches. Personally, I find that the discussion of the single parameter (frequency) case in the paper is very similar to a conventional feedback loop locking scheme in which one could imagine measuring the fluorescence signal and then transforming it appropriately and sending it back to the laser frequency. Even a complex transformation can be done now a days routinely with FPGAs. The main question is: what does the fluorescence signal say about the “hidden” parameters. I think the main contribution of this part of the manuscript is to outline a no-human-in-the-loop determination of this non-linear transfer function but given the centrality of the real time measurement I believe that a much more thorough review of previous approaches examining the signal contained in the fluorescence signal.

In particular because it has over the past two decades, I believe, been the desire of every MOT-graduate student to simply hook the fluorescence signal to an active feedback stabilizer and deliver the more stable atom number output that the authors discuss in the second half of the paper.

Our response #2.3

In our original manuscript we limited ourselves to the presentation of the novel approach and its results, however we agree with Reviewer #2 that a comparison to conventional controllers may be very helpful to the readers. We have added a mention of active feedback controls in the introduction (lines 49-55) including FPGA based approaches and a discussion of fluorescence based control in the literature (lines 369-374).

Reviewer #2, comment #4

- Competing approaches 2: when a significant part of the results is exactly the application to this atom number stabilization using continuous measurement, I find it remarkable that the authors do not compare and contrast their approach to the broader approach of applying QND measurements and feedback. Here in particular it would be relevant to refer to some of the state of the art work eg correcting for a large number of experimental drifts to achieve stabilization down to shot noise level of BEC numbers, <https://journals.aps.org/prl/abstract/10.1103/PhysRevLett.117.073604>

Our response #2.4

We have indeed overlooked the reference mentioned by Reviewer #2. It is clearly a highly relevant work, not only because of its impressive sub-shot noise level of atom number control, but also because it highlights the usefulness of non-destructive imaging beyond the MOT-phase, making it a perfect tool for future work on RL-controlled evaporation cooling. We have added a discussion of this work in the section “Reward function engineering” lines(380-387).

Reviewer #2, comment #5

- *Competing approaches 3: once the issue of determination of potential drift parameters is settled, it would make sense to more clearly discuss the background for the second challenge, the determination of the optimal strategies given these parameters. Here it would be important to give the readers a basic understanding of the relation to basic optimization concepts like gradient based optimization (primarily for simulations) and gradient free approaches such as the very simple remote dressed CRAB approach explored in <https://www.pnas.org/doi/10.1073/pnas.1716869115> for the optimization of ultra cold atoms and then moving on to the ML based approaches briefly mentioned in the introduction. Providing strengths and weaknesses (eg convergence properties, ie number of training/optimization runs necessary) for these approaches will greatly help the reader understand the significance of the current work.*

Our response #2.5

As suggested by Reviewer #2, we have extended the description of existing optimization-based approaches, introducing the cost-function, and the suggested approximation and optimization methods, as well as their limitations (lines 29-48).

Reviewer #2, comment #6

- *Another reason for including a brief discussion about optimization is that it introduces the cost function and thus nuances the statement in the abstract and paper that one of the main novelties is the manipulation of the reward function. I agree that it is a strength of the RL paradigm to engineer the reward function but it is not true that, given the right insights, related results could not be obtained using an optimization technique. As mentioned above, the transfer function may however be complicated and this is where the current approach shines.*

Our response #2.6

As mentioned in the previous response, we have introduced the cost-function and compare it to the reward-function in the revised version of our manuscript (lines 76-79). We further point out that RL is a natural fit when the control of experiment is formulated as a decision making problem (lines 56-63). We argue that when applied at the level of a time step, RL has indisputable advantage over other optimization methods, as it allows for adapted reactions within a single sequence.

Reviewer #2, comment #7

- *Competing approaches 4: one of the main advantages of the ML/RL paradigm compared to other approaches is exactly the “discovery” of novel insights. Compared to eg the discussion in ref 12 about “discovering” redundant parameters I find the physical insights discussions of the current paper could be improved. For instance, it is my impression that in the 1D case of detuning variations, the algorithm “discovers” the importance of maximizing the loading rate and uses a pair of fluorescence measurements to estimate the current rate and then calculate the detuning jump necessary to obtain optimal loading rate. Is this true or is it more complicated than this?*

Our response #2.7

We agree that the potential for the discovery of insights remains to be further explored. In our proof-of-principle experiments we have limited the number of control parameters as well as their range in order to obtain stable training and interpretable control sequences. Nevertheless we have improved discussion of physical insight throughout the manuscript (lines 351-356). As for the “discovery” mentioned by Reviewer #2, we were also expecting something like that. However, it seems that the agent goes beyond merely estimating the current rate, as it is able to react to a sudden jump of the detuning within a single time step. This means it has learned to extract the value of the current detuning offset from a single fluorescence image. We have emphasized this insight in the main text (lines 316-321).

Reviewer #2, comment #8

- *Misc comments: o Page 1 line 44: In providing the list of previous RL papers, since the following paper is to my knowledge the first paper employing the deep RL technique AlphaZero to quantum physics earlier than all of the referenced RL in physics papers (refs 23-227), it is somewhat odd not to cite it: <https://www.nature.com/articles/s41534-019-0241-0>*

Our response #2.8

We thank Reviewer #2 for drawing our attention to this work. We have added it to the list of references in the introduction.

Reviewer #2, comment #9

o Page 4 line 249: not clear how the variations in control policy are realized? is it by running a number of training events each w 20k episodes, or by evaluating the same converged agent/policy on a number of experimental realizations? i assume the former, but it is not clearly described how many times this was run. isnt the lack of variation a sign of the simplici of the problem rather than proof that the "control policy is remarkably stable"?

Our response #2.9

The variations referred to by Reviewer #2 are realized by evaluating the same agent on a number of experimental realizations. We provide the number of realizations (200) in the subsection "Evaluating the agents performance" (line 232). We also state in the caption of Fig. 2 a-c that the displayed control parameter sequences were obtained from the same agent. As of the lack of variations, indeed, they are more a sign of a stable MOT. We have omitted the statement about the stability of the control policy. The policy's robustness is primarily demonstrated by its effective response to the abrupt shift of the detuning in the middle of the episode.

Reviewer #2, comment #10

o Page 4 line 250: again it is not clear how the error bars in fig 2e are obtained.

Our response #2.10

We state in the caption to Fig. 2e (now **d**) that the error bars correspond to the standard deviation (of the respective 200 evaluations).

Reviewer #2, comment #11

o Page 5 line 314: not so surprising, given there is more temporal freedom. more interesting is why this gain only happens for negative detunings. what is the physical insight gained from this?

Our response #2.11

We thank Reviewer #2 for drawing our attention to the gain in Fig. 2 **e** (now **d**). We have removed the datapoints corresponding to the experiments with 2 control parameters from Fig. 2**e**, because in the process of revision we have determined, that the data were normalized with respect to a non-optimal reference sequence. We agree, that additional degrees of freedom should eventually lead to better performance. At the same time, the parameter space is increased as well, so it is not obvious that the agent should be able to converge to any meaningful policy at all. By allowing the agent to additionally act on the gradient we have shown that the training remains stable. Furthermore, we hypothesize (lines 351-353) that the agent has learned to better detect the fluorescence signal in the first time steps of an episode by increasing the gradient to its maximum.

Reviewer #2, comment #12

o Page 6, line 362: again, the comparison to other active control strategies has no been made clear. in particular, it is not clear how this method would compare in performance to either an active lock to the loading rate (adjusted for detuning dependent flourescenece rate) or even simpler a tabulated loading rate vs detuning single step look up. such implementtions are somewht complex to implement using analog control electronics but easy to implemnt on eg an FPGA.

Our response #2.12

We agree that it is possible to implement a similar controller on an FPGA. However, in the mentioned subsection we would like to convince the reader that, in general, the task of the agent can be customized by adapting the reward-function, i.e. without the implementation of hand-crafted control algorithms.

Reviewer #3

The authors use reinforcement learning to control a MOT. They show how the system can be trained both from real data and in-silico and demonstrate a robust performance of the hardware + software loop. The main difference from previous implementations of similar ideas is that there is not pre-determined experimental sequence, instead the network adaptively controls the experiment based on intermediate non-destructive measurements.

The work is nicely done and I appreciate that the authors have made available both the data, simulations, and code for other interested parties to reproduce their results. I generally find that the work presented here is in line with the recent trend of using neural networks for tasks that do not necessarily benefit from them. Although the NN part of the work is a tour-de-force the physics part remains largely unconvincing. The authors control one or two parameters of the system only and cannot reproduce/discover standard techniques like cMOT because of the slow fps.

Our response #3.0

We thank the Reviewer #3 for his thorough evaluation of our work and his comments on the manuscript. Throughout the manuscript, they helped us to eliminate ambiguities and to be more precise on specific details and conclusions.

Major comments.

Reviewer #3, comment #1

- Reinforcement learning algorithms are notoriously known for the amount of samples they need for training. That might be fine for a MOT sequence but I suspect it will start becoming a problem in a full BEC cycle. Already the training of a MOT takes up to 2 days. The authors also never discuss how long for that training remains valid for. In the end of the day all this learning is contingent to the experimental stability and reproducibility. How stable do conditions in the lab need to be for the NN to keep working? How long does it actually keep working for before it needs to be retrained.

Our response #3.1

We thank Reviewer #3 for bringing up the importance of the duration of the validity of a training. It is true that lab conditions are often subject to slow changes, which might lead to failure of the trained agent. We have extended our manuscripts with results showing how RL is capable of dealing with long-term instabilities via continued training (lines 514-583). In short, the agent can be further trained on observables and rewards generated during deployment after the exploration is disabled (i.e. during the agents use in normal experimental control). Any changing conditions will be reflected in these new data and thus will allow the agent to follow the environment. We have also provided additional details of how experienced time steps are used in the training of the agent in the Methods section (lines 791-798, 882-893), that should enable better understanding of the underlying mechanisms. These arguments also hold if the agent would be trained to produce a BEC.

Furthermore, another advantage of RL is that it is easily possible to provide the agents with additional observables such as the power of the lasers or the temperature in the lab, reflecting the changing lab conditions. Ongoing experiments are currently focusing on this subject.

Reviewer #3, comment #2

- I am assuming that the authors are after doing BEC physics ultimately (please forgive me if this assumption does not hold) however, experientially the MOT atom number is only weakly correlated with the BEC atom number. The later can be and has been stabilised to the point of being limited by Poissonian fluctuations in *Phys. Rev. Lett.* 122, 163601 and related work from the same group using a simple feedback loop with any NN.

Our response #3.2

On the one hand side, extending our research to creation of degenerate gases with the help of RL is indeed a fascinating perspective. For this, one would resort to non-destructive imaging techniques such as Faraday imaging as used in the reference mentioned by Reviewer #2 (Phys. Rev. Lett. 122, 163601). In such a case, the reward would most likely be a function of phase space density and number of atoms in the condensate, without explicitly optimizing the MOT atom number. We have also added a brief discussion of feedback loops compared to our approach in the main text (lines 49-55, 375-387).

On the other hand side, there are plenty of applications such as atomic clocks and quantum repeaters, which are based on atomic samples in a MOT, that could profit from a reliable, RL-controlled MOT.

As stated in the outlook section, another promising area of application of RL algorithms is the MOT of non-alkali atoms and even molecules, which require larger numbers of lasers and more complex sequences.

Reviewer #3, comment #3

- The authors only control one or two parameters. Even in a MOT these are too few. There is power and detuning of cooling and repumper lasers, polarisations, control of beam imbalance, etc. I'm guessing adding more parameters would make the training time prohibitively long. It might be more interesting to control the shape of the MOT however that the atom number in order to make subsequent stages more efficient.

Our response #3.3

We have observed that extending the control to 2 parameters did not dramatically increase the time for convergence. In robotics applications the number of controlled parameters (degrees of freedom) is typically on the order of 10. Therefore, the right exploration strategy is the key to efficient learning. Recently, DEP-RL (Schumacher et al) has shown impressive results on challenging musculoskeletal systems, with over 50 controllable muscle actuators. We fully agree that the control of the shape and position of the MOT is essential for subsequent stages such as an efficient transfer of atoms into conservative potentials. It will be interesting to study the end-to-end policies of an agent that is tasked to maximize the initial phase space density in an optical dipole trap.

Reviewer #3, comment #4

In any case, due to the slow fps, some standard techniques like cMOT will always be out-of-reach for the NN. Moreover, the NN cannot detect any fluctuations faster than 60Hz if they are synchronous with its sampling rate.

Our response #3.4

We agree with Reviewer #3 that with our reported time step duration of 60 ms, techniques that rely on short ramps and hold times are not feasible. However, the duration can be significantly reduced using a more sensitive camera and adapted optics. In our experiments we have limited the frame rate out of convenience of using a regular desktop PC, thereby lowering the technical threshold for an easy access to RL for the cold atoms community. From technical point of view, commercial CMOS cameras are capable of capturing hundreds of frames per second. In a recent publication (<https://doi.org/10.1038/s41467-023-42901-3>), agent interaction on sub-microsecond timescale is demonstrated, allowing for feedback action to a qubit system. It is also conceivable to give the agent the control over the duration of each time step.

Reviewer #3, comment #5

- I find the presentation of the NN as completely sequence-agnostic somewhat misleading. It's only due to the MOT being a stable object with infinite lifetime and a human having made a number of decisions about all the other parameters that make it possible and about the limits of the parameters the NN explores. The distinction of those decisions not involving time is a pretty fine one

Our response #3.5

We agree with Reviewer #3 that a significant portion of human-expert knowledge is required to prepare a MOT for operation by a RL agent. In lines 471-474 of the original submission we explicitly state that

the limits imposed on the control parameters were put in place in order to facilitate convergence and obtain comprehensible control strategies, since this is, to our knowledge, the first implementation of RL on such a system. Nevertheless, it is important to note that the agent is completely agnostic of what the pixels of the fluorescence images mean, how the images are affected by the controlled parameter, and how the reward is calculated. Therefore it is quite remarkable that we observe learning of meaningful control strategies. Furthermore, RL is rarely applied to any real-world system, especially outside of the traditional RL community. For this reason, we firmly believe that our approach is sound and will enable many other researchers to explore RL in their respective area.

Reviewer #3, comment #6

- *The authors miss a key reference, A J Barker et al 2020 Mach. Learn.: Sci. Technol. 1 015007, where various NN techniques were explored for BEC optimisation. This somewhat diminishes the novelty of the work.*

Our response #3.6

The undoubtedly important work by Barker et al was cited (in the original manuscript) as Ref. 13. In the revised manuscript we have extended the discussion of state-of-the-art machine learning techniques in the context of BEC optimization (lines 29-48) in order to better highlight the fundamental difference to reinforcement learning and thereby underline the novelty of our approach (lines 56-81). We have also added a paragraph (lines 446-451) discussing the engineering of the reward function compared to tailoring of the cost function as proposed by Barker et al.

Reviewer #3, comment #7

- *During the training phase, the authors put artificial offsets to make the NN more robust. Changes like these are synchronous and are passed to the next timestep. If the field was to have a large spike and return to the same value the algorithm would be tricked. Why not change some real environmental parameters to emulate actual hardware variation?*

Our response #3.7

We agree with Reviewer #3, that the offsets during training are applied synchronously to the time steps and are constant during a single episode. However, during evaluation, sudden shifts are applied, as shown in Fig.2c, where the initial offset is discontinuous. These shifts can be considered an emulation of hardware variation. The capability of the agent to compensate for these shifts within a single time step demonstrates that the policy is robust against asynchronous variations. Nevertheless, it is clear, that perturbations at frequencies above the control bandwidth would present a challenge for any type of feedback controller.

Reviewer #3, comment #8

Minor comments.

- *Episodes seems to be coach-specific jargon. Why not epochs?*

Our response #3.8

We employ the term "episode", because it corresponds to a sequence of states and is very common in RL literature, whereas "epochs" typically denote a number of optimization steps (weight updates) of a neural networks.

Reviewer #3, comment #9

- *It wasn't clear to me in what sense are the fluorescence images not part of the cost function. Why are they being taken every 60ms then?*

Our response #3.9

The fluorescence images are taken every 60ms and serve as input for the agent, they represent the state based on which the agent decided which action to take. We have highlighted the role of the fluorescence

images in the revised version of manuscripts in lines 92-95,163-167,603-608. Furthermore, we have added an explanation on how reward function differs from cost functions which are used in other works (lines 76-79).

In our experiments the feedback control uses observations (i.e. fluorescence images), while the training is carried out using the reward (i.e. calculated from absorption images), as illustrated by the new Fig. 1b. In the revised manuscript, we have pointed out that the control of the MOT and the training of the agent are two separate processes (lines 92-95, 173-175, 188-191).

In general, in RL the input (observation) does not necessarily go directly into the reward function.

Reviewer #3, comment #10

- Is the processing time part of the 60ms cycle or is there an additional dead time as well?

Our response #3.10

The processing of the fluorescence images for the computation of the next action takes place during the 60ms of a time step. The training takes place after each episode, while the experiment resets.

Reviewer #3, comment #11

- Panels in figures 2 and 3 are referenced out of order which makes for a confusing read.

Our response #3.11

Generally, our main objective in composing the figures was to make them self-contained and mostly comprehensible without reading the main text. As suggested by Reviewer #3, we have swapped panels **d** and **e** in Figure 2 to match the order in which they are referred to in the main text.

Reviewer #3, comment #12

- I would be interested to see MOT loading curves for the atom-number-fixing experiments. Has the NN understood to drive the MOT into saturation and hence reduce the number fluctuations or is stopping the loading on a positive slope?

Our response #3.12

We thank Reviewer #3 for this comment, as it let us notice that we omitted to explicitly state that in the presented experiments the number of atoms does not reach saturation within the duration of an episode. We have added this statement in line 134. Indeed, driving the MOT into saturation and subsequently reducing the number of atoms would be a natural approach. Instead, as we discuss in the text, the agent controls the loading rate via the detuning distance to the optimal detuning. The agent reduces the number fluctuations by actively adapting the loading rate in each time step and stopping the loading depending on the current fluorescence image. Thanks to comments of Reviewer #2 and 3, we have added a passage comparing our approach to works that prepare samples with fixed atom number by ejection of excessive atoms (lines 375-387). With this said, we believe that showing the fluorescence curve over the course of an episode ('MOT loading curves') would not be beneficial for the understanding of the learned policy.

Reviewer #3, comment #13

- 60ms sampling is not the same as real-time feedback and I believe the language should be changed in the manuscript. It might be faster than typical experimental cycles in atomic physics but at the same time it's too slow for discovering sharp-change sequences.

Our response #3.13

We use the term "real-time feedback" in order to emphasize the fact that decisions on actions are taken during the sequence and as a function of the observations. The experimental cycle is comprised of 25 time steps, resulting in an experimental cycle of 1.5s, which is rather typical for MOT experiments. We, too,

believe, that increasing the sampling rate opens the potential for discovery of new control sequences and are currently preparing corresponding experiments.

Reviewer #3, comment #14

- It's odd that capturing the total fluorescence didn't work well. Why? What feature of the fluorescence images is used in the end? Also I thought they weren't used in the reward.

Our response #3.14

It is correct that the fluorescence images are not used in the reward. However, they are used during the sequence, i.e. in each step the agent extracts useful features from the fluorescence images and determines the next value of the control parameter. The summation of all pixels of a fluorescence image in order to obtain the total fluorescence erases spatial information contained therein. We believe that this spatial information is used by the agent to estimate the current detuning value and thus is necessary for compensation of offsets. We have added a passage highlighting the fact that the agent is able to compensate a sudden detuning shift within a single time step (lines 316-318).

REVIEWERS' COMMENTS

Reviewer #1 (Remarks to the Author):

In the revised version of the manuscript, Reinschmidt et al have done an excellent job in improving the manuscript following all reviewers' comments. It reads excellently, the fine details of both atomic physics and ML aspects are nicely distilled (enough for specialists, not too much for novices). It is clearly a thorough paper bringing new aspects to the field. I appreciate the effort in extracting some physics from the optimal strategies. As common with such works, especially for a broad audience journal, I would have preferred more, novel, physics insights.

It is definitely worth publishing, and it should have a large impact in the atomic physics community. I'm unsure whether Nature Communications is the proper venue for publication, and I live that decision to the editors.

Minor comments:

1- Line 267, I believe the reference should be to Fig. 1c.

2- Since the authors first submitted their manuscript, at least one other group published results on ML in atomic physics experiments: <https://doi.org/10.1088/2632-2153/ad1437>
From scrolling through it, the present manuscript by Reinschmidt et al is complementary, with the significant addition of live "adjustment". And it was posted before on the arxiv. Still, for completeness, the authors should probably cite that publication.

Reviewer #2 (Remarks to the Author):

The authors have addressed all my concerns to satisfaction and i am happy to recommend publication.

Reviewer #3 (Remarks to the Author):

I believe the authors improved the manuscript substantially with the help of the comments from the referees.

The difference between previous NN techniques and RL is now clearly discussed, the description of all the steps in the method is much clearer, and the added section on long-term stability is important to assess the performance of the agent.

I still find that the long training times, the need for software-to-hardware training, the general simplicity of the problem, and the stability of the hardware make the RL demonstration a bit lacklustre.

However, I think the work is novel and could potentially help with simple devices that are based on a MOT, and points the way to how RL techniques could be integrated in such systems and how they would fare in real laboratory or real life conditions.

The minor comments by Reviewer #1 have been taken into account:

1- Line 267, I believe the reference should be to Fig. 1c.

-> This is correct. We have corrected this reference in the manuscript.

2- Since the authors first submitted their manuscript, at least one other group published results on ML in atomic physics experiments:

<https://doi.org/10.1088/2632-2153/ad1437>

From scrolling through it, the present manuscript by Reinschmidt et al is complementary, with the significant addition of live "adjustment". And it was posted before on the arxiv. Still, for completeness, the authors should probably cite that publication.

-> We agree that the mentioned work is complementary to our study. We have added the citation to the manuscript in a note at the end of the discussion.